

# Spiking neuromorphic chip learns entangled quantum states

**Stefanie Czischek[1,2]⋆, Andreas Baumbach[1,3], Sebastian Billaudelle[1],**
**Benjamin Cramer[1], Lukas Kades[4], Jan M. Pawlowski[4], Markus K Oberthaler[1],**
**Johannes Schemmel[1], Mihai A. Petrovici[3,1], Thomas Gasenzer[1,4], Martin Gärttner[1,4,5]**

**1** Kirchhoff-Institut für Physik, Ruprecht-Karls-Universität Heidelberg,
Im Neuenheimer Feld 227, 69120 Heidelberg, Germany
**2** Department of Physics and Astronomy, University of Waterloo, Ontario, N2L 3G1, Canada
**3** Department of Physiology, University of Bern, 3012 Bern, Switzerland
**4** Institut für Theoretische Physik, Ruprecht-Karls-Universität Heidelberg,
Philosophenweg 16, 69120 Heidelberg, Germany
**5** Physikalisches Institut, Universität Heidelberg,
Im Neuenheimer Feld 226, 69120 Heidelberg, Germany

⋆ sczischek@uwaterloo.ca

## Abstract

The approximation of quantum states with artificial neural networks has gained a lot of attention during the last years. Meanwhile, analog neuromorphic chips, inspired by structural and dynamical properties of the biological brain, show a high energy efficiency in running artificial neural-network architectures for the profit of generative applications. This encourages employing such hardware systems as platforms for simulations of quantum systems. Here we report on the realization of a prototype using the latest spike-based BrainScaleS hardware allowing us to represent few-qubit maximally entangled quantum states with high fidelities. Bell correlations of pure and mixed two-qubit states are well captured by the analog hardware, demonstrating an important building block for simulating quantum systems with spiking neuromorphic chips.



# 1   Introduction

As von-Neumann computers are rapidly approaching fundamental physical limitations of conventional semiconductor technology, a number of alternative computing architectures are currently being explored. Among them, neuromorphic devices [1,2], which take inspiration from the way the human brain works, hold promise of having a wide range of applications, in particular in machine learning and artificial intelligence [3–11]. Here we focus on using them as a sampling device to emulate measurement outcomes in quantum physics [12], which are inherently probabilistic in nature. The BrainScaleS neuromorphic system [11] is ideally suited for this task. The accelerated analog circuit dynamics and the inherently parallel nature of the neuromorphic substrate enable a rapid generation of samples which carries the potential of scaling benefits as compared to von-Neumann devices (App. A).

We use neuronal spikes (action potentials) to mark transitions between discrete states and thereby effectively carry out the sampling process. The all-or-nothing nature of spikes represents a blessing in disguise. On the one hand, it does have an apparent drawback by making the computation of gradients – and thus, training – more demanding than in classical deep neural networks [2]. On the other hand, it also allows us to use a spiking neuromorphic substrate in the first place, the speed-up of which we harness for efficient Hebbian learning [13].

Since any quantum state can be mapped to a probability distribution [14,15], it can, in turn, be represented using networks of leaky integrate-and-fire (LIF) neurons [16–18]. Here, we use the BrainScaleS-2 chip [11] as a physical substrate to emulate such networks. This mixed-signal neuromorphic platform is centered around an analog core: neuro-synaptic states are represented as voltages and currents in integrated electronic circuits and evolve in continuous time. Its configurable connectivity of neurons allows us to explore various different network topologies, including shallow, as well as deep and densely connected ones. With this substrate, we demonstrate an approximate representation of quantum states with classical spiking neural networks that is sufficiently precise for encoding states with genuine quantum correlations.

# 2   Neuromorphic encoding of quantum states

In classical machine learning, generative models based on artificial neural networks are used to encode and sample from probability distributions [13]. Similarly, spiking neural networks can be viewed as approximating Markov-chain Monte-Carlo sampling, albeit with dynamics that differ fundamentally from standard statistical methods [19]. Here, we encode quantum states

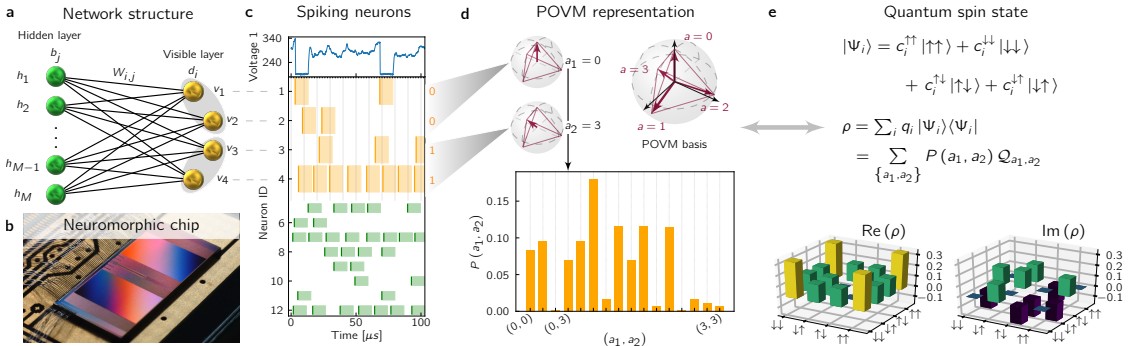

Figure 1: **Neuromorphic representation of quantum states. a**, Two-layer spiking network architecture with weight parameters $W_{i,j}$ between the visible (orange) and hidden (green) neurons and biases $d_i$ ($b_j$) for the binary visible (hidden) neurons. **b**, Photograph of the BrainScaleS-2 chip used as a substrate for the experiments in this work. **c**, Dynamical evolution of the spiking network. Upper panel: membrane potential evolution of a single LIF neuron integrating synaptic input. Whenever the potential crosses a threshold a spike is generated and the potential is clamped to prevent immediate refiring (refractory period). Lower panels: Spikes (solid lines) for 4 visible (orange) and 8 hidden (green) neurons with associated $z = 1$ time frames (shaded regions). The network state is observed periodically (gray lines showing only every fifth observation time for visibility reasons). Each observation results in a binary vector corresponding to a sample drawn from the underlying distribution. **d**, The 4-state positive-operator-valued measure (POVM) representation of a qubit state can be encoded by a pair of visible neurons. A combination of $N$ such neuron pairs thus serves to represent an $N$-qubit system. The frequency of occurrence of neuron configurations drawn from a trained network encodes the POVM probability distribution of a quantum state (lower panel). **e**, Any quantum state can be represented as a density matrix $\rho$, which can be a statistical mixture of states $|\Psi_i\rangle$. For the example of two qubits shown here, the complex-valued entries of $\rho$ can be reconstructed linearly from the sampled probabilities $P(a_1, a_2)$. For the definition of the operators $\mathcal{Q}_{a_1,a_2}$, see App. C.

using the hierarchical network architecture illustrated in Fig. 1a. The network consists of $N$ visible and $M$ hidden leaky integrate-and-fire (LIF) neurons arranged in a bipartite graph with a symmetric connectivity matrix. Such a network can be tuned to approximate the probability of the visible neurons to be in state $\boldsymbol{v} = (v_1, \ldots, v_N)$, $v_i \in \{0, 1\}$, as the marginal

$$p(\boldsymbol{v}; \mathcal{W}) = \frac{1}{Z(\mathcal{W})} \sum_{\{\boldsymbol{h}\}} \exp\left[-E(\boldsymbol{v}, \boldsymbol{h}; \mathcal{W})\right], \tag{1}$$

over all hidden states $\boldsymbol{h} = (h_1, \ldots, h_M)$, where $h_j \in \{0, 1\}$, of the joint Boltzmann distribution $p(\boldsymbol{v}, \boldsymbol{h}; \mathcal{W}) = \exp\left[-E(\boldsymbol{v}, \boldsymbol{h}; \mathcal{W})\right]$ [19]. The network energy $E(\boldsymbol{v}, \boldsymbol{h}; \mathcal{W}) = -\sum_{i,j} v_i W_{i,j} h_j - \sum_i v_i d_i - \sum_j h_j b_j$ depends on the set of network parameters $\mathcal{W} = (W, \boldsymbol{b}, \boldsymbol{d})$ including the weights $W_{i,j}$ and biases $b_j$ and $d_i$. The partition sum $Z(\mathcal{W}) = \sum_{\{\boldsymbol{v}, \boldsymbol{h}\}} p(\boldsymbol{v}, \boldsymbol{h}; \mathcal{W})$ ensures normalization.

The BrainScaleS-2 system, depicted in Fig. 1b, features 512 LIF neuron circuits interacting through a configurable weight matrix [11]. Similar to biological neurons in the human brain, LIF neurons communicate via spikes. Each neuron can be viewed as a capacitor integrating the currents it receives from its synaptic inputs to generate a membrane potential. Whenever this

membrane potential crosses a threshold from below, the neuron sends a spike to the synaptic inputs of its efferent partners (Fig. 1c, top panel). After sending a spike, the neuron is set to an inactive state, in which no additional spike can be triggered for a certain time, referred to as the refractory period $\tau_{\text{ref}}$. In the spike-based sampling framework, neurons in this refractory state encode the state $z = 1$, and $z = 0$ at all other times (Fig. 1c, lower panel). The stochasticity required for sampling is induced by adding a random component to the generation of spikes; for LIF networks, this can be ensured by sufficiently noisy membrane potentials [16, 18]. To this end, we used on-chip sources to inject pseudo-Poisson spike trains into the network (see App. A).

As an experimental result, the BrainScaleS-2 chip returns a list of all spike times and associated neuron IDs. This information is sufficient to reconstruct the network state at any point in time. We estimated the distribution sampled by the network by observing its state at regular intervals, as visualized in Fig. 1c. To ensure an optimal estimate, the observation frequency needs to be at least $(\tau_{\text{ref}}/2)^{-1}$ (see App. A). For our analysis, we used $(\tau_{\text{ref}}/5)^{-1}$, thereby guaranteeing a large safety margin. The resulting binary configurations are collected in a histogram as shown in Fig. 1d.

A pure quantum state is described by a vector in Hilbert space and can be represented by a hermitian density matrix with complex entries. Density matrices can also encode mixed states and thus account for a possible coupling to an environment, which is relevant for a realistic description of experiments. Fig. 1e shows an example of a density matrix for a system of two spin-$1/2$ degrees of freedom (qubits) corresponding to a Hilbert-space dimension $d = 4$. The corresponding probability distribution which we encode in our network is obtained from a so-called tomographically complete measurement [14]. Such a measurement has $d^2$ possible outcomes. Mathematically, these outcomes are represented by a set of operators $\{M_{\boldsymbol{a}}\}_{\boldsymbol{a}}$, forming a so-called positive-operator-valued measure (POVM). The density matrix can be reconstructed uniquely as $\rho = \sum_{\{\boldsymbol{a}\}} P(\boldsymbol{a})\mathcal{Q}_{\boldsymbol{a}}$ from the probabilities $P(\boldsymbol{a}) = \text{Tr}[\rho M_{\boldsymbol{a}}]$ for obtaining outcome $\boldsymbol{a}$ according to Born's rule. The operators $\mathcal{Q}_{\boldsymbol{a}}$ are given by $\mathcal{Q}_{\boldsymbol{a}} = \sum_{\{\boldsymbol{a}'\}} T_{\boldsymbol{a},\boldsymbol{a}'}^{-1} M_{\boldsymbol{a}'}$, with $T_{\boldsymbol{a},\boldsymbol{a}'} = \text{Tr}[M_{\boldsymbol{a}} M_{\boldsymbol{a}'}]$ [15]. Hence, any density matrix $\rho$ can be mapped to a probability distribution $P(\boldsymbol{a})$, and the information contained in the quantum state can be retrieved from that distribution. In our two-qubit example (Fig. 1d) we chose $M_{\boldsymbol{a}} = M_{a_1} \otimes M_{a_2}$, where $M_{a_i}$ ($a_i = 0, \ldots, 3$) are projection operators onto the single-qubit states represented as the four corners of a tetrahedron on the Bloch sphere. As each $a_i$ can take four different values, the encoding of the probabilities $P(\boldsymbol{a})$ by a spiking network is realized by representing each qubit by a pair of binary neurons in the visible layer (cf. gray shadings in Fig. 1a). This results in the distribution $p^*(\boldsymbol{v})$ over the visible neurons (see App. C).

To approximate $p^*(\boldsymbol{v})$ through spike-based sampling, the parameters of the spiking network were adjusted in an iterative training procedure. We used the Kullback-Leibler divergence

$$D_{\text{KL}}(p^* \| p) = \sum_{\{\boldsymbol{v}\}} p^*(\boldsymbol{v}) \ln\left[\frac{p^*(\boldsymbol{v})}{p(\boldsymbol{v}; \mathcal{W})}\right], \tag{2}$$

to measure the quality of the sampled marginal $p(\boldsymbol{v}; \mathcal{W})$. In each training epoch, the synaptic weights were updated along the gradient of the $D_{\text{KL}}$ (see App. D):

$$\Delta W_{i,j} \propto \left\langle v_i h_j \right\rangle_{\text{target}} - \left\langle v_i h_j \right\rangle_{\text{model}}, \tag{3}$$

which is derived assuming that the distribution $p(\boldsymbol{v}; \mathcal{W})$ is given by Eq. (1). While the dynamical behavior of the spiking hardware approximates this probability distribution, the exact relation between the network parameters and the encoded distribution cannot be given in a closed form [18]. Instead, pairwise correlations $\left\langle v_i h_j \right\rangle_{\text{model}}$ in the network were measured

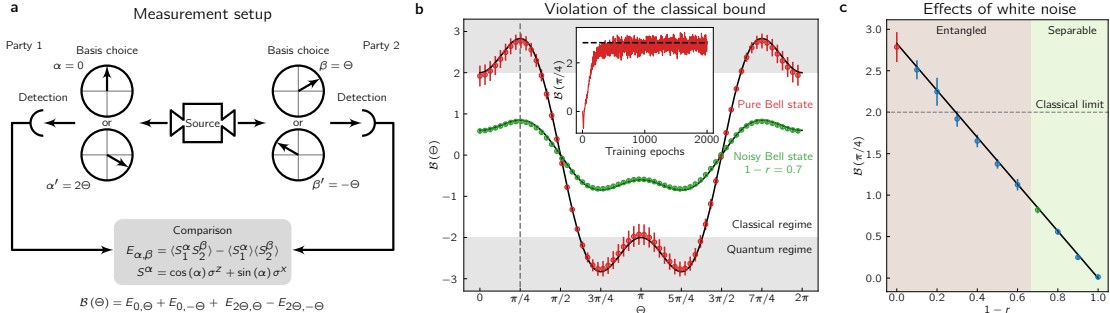

Figure 2: **Encoding Bell states and Werner states. a,** Illustration of a typical Bell-test scenario. Two correlated qubits emerging from a source are distributed between two parties. Each of the parties is allowed to choose between two different measurements each characterized by a single common angle $\Theta$. The measurement outcomes indicate genuine quantum correlations if the combination $\mathcal{B}(\Theta)$ of the correlations violates the inequality $|\mathcal{B}(\Theta)| \leq 2$ obeyed by classical states. **b,** Observable $\mathcal{B}(\Theta)$ evaluated on the learned encoding of the Bell state $\rho_B = |\Psi^+\rangle\langle\Psi^+|$ on the neuromorphic hardware, with $M = 20$ hidden neurons. Red symbols depict the observable for different angles $\Theta$, averaged over the last 200 training epochs, where errorbars here and in the following denote the standard deviation. Note that these data points have been obtained from the same trained network and the same set of neuron states sampled from it by evaluating the observable for different angles $\Theta$ on this sample set. Werner states $\rho_W = r\rho_B + (1-r)\,\mathbb{1}/4$ are obtained by adding white noise to the pure Bell state. Green points correspond to $r = 0.3$. In both cases, the data capture the exact values (black lines) well, including the violation of the classical bound in the pure case $r = 1$. The inset shows the evolution of the Bell-correlation witness $\mathcal{B}(\Theta = \pi/4)$ during training (red line) and the convergence towards the expected value (black dashed line). **c,** Bell-correlation witness $\mathcal{B}(\Theta = \pi/4)$ for a Werner state as a function of the noise strength $1-r$. The exact solution (black line) is captured well for both entangled and separable states.

from the sampled distribution $p(\boldsymbol{v}, \boldsymbol{h}; \mathcal{W})$. Target correlations $\langle v_i h_j \rangle_{\text{target}}$ were also obtained from the sampled distribution by renormalization to the target marginal distribution:

$$\langle v_i h_j \rangle_{\text{target}} = \left\langle \frac{p^*(\boldsymbol{v})}{p(\boldsymbol{v}; \mathcal{W})} v_i h_j \right\rangle_{p(\boldsymbol{v}, \boldsymbol{h}; \mathcal{W})}. \tag{4}$$

A similar scheme was used for the neuronal biases $b_j$ and $d_i$. The performance characteristics of the neuromorphic hardware make computing additional samples cheap compared to reconfiguration and reinitialization. Hence we can take into account the complete sampled distribution for the update calculation, rather than relying on few-sample approximations as in contrastive divergence [13]. This enables a much better estimation of the $D_{\text{KL}}$ gradient and does not rely on layer-wise conditional independence, allowing the exploration of network topologies other than bipartite graphs. See App. D for an extended discussion of the learning scheme.

## 3  Encoding an entangled Bell state

To demonstrate that a spiking neural network can learn to represent entangled quantum states we focus on a maximally entangled two-qubit state, the Bell state $|\Psi^+\rangle = (|\uparrow\uparrow\rangle + |\downarrow\downarrow\rangle)/\sqrt{2}$.

This state is a prototypical example exhibiting quantum mechanical correlations [20, 21]. We trained a network of four visible and 20 hidden neurons to encode the POVM probability distribution corresponding to $\rho_B = |\Psi^+\rangle\langle\Psi^+|$. For calculating the weight updates in each epoch of the training procedure, as well as for evaluating expectation values, we drew 125000 samples of neuron states. This number is sufficient for the saturation of the $D_{KL}$ as can be seen in Fig. 3b and was used for all experiments, if not specified otherwise.

To characterize the learned quantum state, we used the observable $\mathcal{B}(\Theta)$, which can signal genuine quantum correlations and is experimentally accessible via measurements as illustrated and defined in Fig. 2a: The two qubits are distributed to two parties who independently perform one of two possible measurements on their respective qubit. We choose the standard parametrization of the different measurements by a single angle $\Theta$. For a Bell state this procedure yields correlations violating the inequality $|\mathcal{B}(\Theta)| \leq 2$, which is obeyed by classical systems [21]. At $\Theta = \pi/4$ this inequality is maximally violated for the Bell state $\rho_B$ and thus yields an experimentally accessible witness for Bell correlations [20, 22].

The correlations in the quantum states encoded as probability distributions by the trained spiking network clearly exceed the classicality bound $|\mathcal{B}(\Theta)| = 2$ (red points in Fig. 2b) and are in agreement with their exact $\Theta$-dependence (black line). The inset shows how the Bell correlation witness $\mathcal{B}(\Theta = \pi/4)$ develops during the training, converging after less than 1000 iterations.

To illustrate the generality of our neuromorphic encoding scheme we consider mixed quantum states by adding white noise to the pure Bell state resulting in the Werner state $\rho_W = r\rho_B + (1-r)\mathbb{1}/4$ with noise strength $0 \leq 1-r \leq 1$ [23]. Increasing the noise reduces $|\mathcal{B}(\Theta)|$ and eventually confines it within the classical regime (cf. green data in Fig. 2b). For $1-r > 1/\sqrt{2}$ the Bell correlation witness fails to detect entanglement, and for $1-r > 2/3$ the state becomes separable (unentangled). The resulting mixed states are faithfully represented by our system for any value of $r$ as shown in Fig. 2c. The fluctuations in the experimental data decrease with increasing noise contribution, allowing a more accurate learning of mixed states. This counterintuitive effect is due to additional noise leading to an increase in entropy, which is synonymous with sampling from more uniform distributions. These, in turn, are realized by weaker weights, thus decreasing the influence of imperfect synaptic interactions in the neuromorphic substrate.

## 4 Learning performance

We analyzed in detail the convergence of the learning algorithm using the classical Kullback-Leibler divergence $D_{KL}$ as defined in Eq. (2). In addition, we use the quantum fidelity

$$\mathcal{F}(\rho_B, \rho_N) = \text{Tr}\left[ \sqrt{\sqrt{\rho_B}\rho_N\sqrt{\rho_B}} \right], \tag{5}$$

to quantify the distance between the target state $\rho_B$ and the network-encoded state $\rho_N$, which, for pure states, reduces to the state overlap. As shown in Fig. 3a, the learning converges after 1000 training epochs. Increasing the number of hidden neurons we find that the fidelity reaches $\approx 98\%$ (correspondingly $D_{KL} \lesssim 10^{-2}$) for $M \gtrsim 20$ hidden neurons. The limited reachable fidelity is a result of many different factors of the physical implementation of the spiking neural network on the BrainScaleS-2 platform. The synaptic connections are implemented with 6-bit resolution, limiting the achievable precision of approximating the probability distribution. Also, uncontrolled environmental changes such as temperature variations or host-to-system effects influence the performance of the hardware. This manifests in the jumps of fidelity occurring during learning, as well as in strong noise in the fidelity after the learning process has saturated, as can be seen in Fig. 3a. These instabilities exceed the anticipated noise

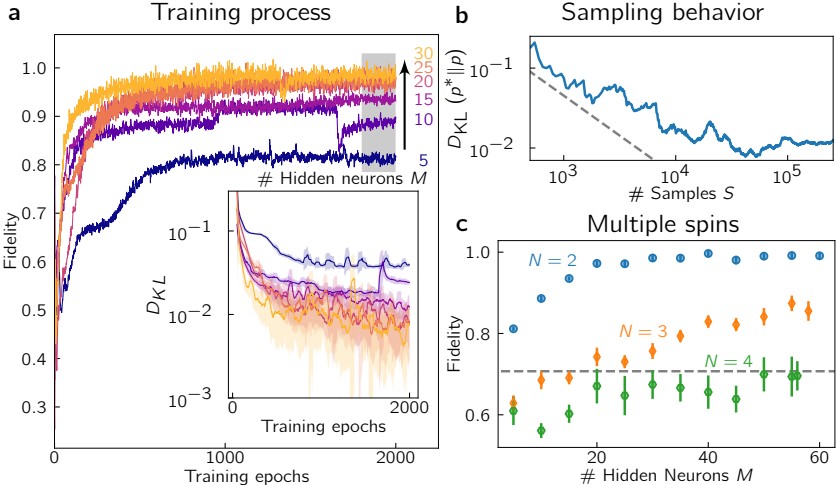

Figure 3: **Training performance. a**, Dynamics of the learning procedure for the pure Bell state $\rho_B$. The quality of the network-encoded state is measured by the quantum fidelity, Eq. (5) (main frame), and by the Kullback-Leibler divergence, Eq. (2) (inset), for different numbers of hidden neurons. For better visibility, the running average over 50 epochs is shown in the inset as solid lines, with the shaded areas indicating the corresponding standard deviation. **b**, Kullback-Leibler divergence in a fixed trained network with $M = 20$ hidden neurons as a function of the number of samples drawn. The dashed line shows the expected trend for exact sampling from the target distribution. **c**, Quantum fidelity as a function of the number of hidden neurons for GHZ states $|\Psi\rangle = (|\uparrow\rangle^{\otimes N} + |\downarrow\rangle^{\otimes N})/\sqrt{2}$ with $N = 2$ (Bell state), 3, and 4 qubits. We show the averages over 200 training epochs after convergence (gray shaded area in **a**). The dashed line shows the bound for genuine $N$-partite entanglement.

level due to finite sample statistics used for evaluating observables and calculating gradients in each epoch. These factors degrade the correspondence between the model assumption underlying the employed learning rule and the actual dynamics of the hardware. Many of the issues mentioned above can be resolved in future hardware generations.

To ensure that the learning performance is not limited by finite sample statistics, we evaluated the Kullback-Leibler divergence as a function of the number of samples in a trained network with fixed network parameters. Figure 3b shows the expected convergence towards a minimum value determined by the quality with which the spiking network approximates the POVM distribution. Typically, for $>10^5$ samples the statistical error is negligible compared to the errors due to hardware noise and limited representational power of the network, causing the saturation of the DKL observed in Fig. 3b. This justifies our choice of training with 125000 samples per epoch.

Having demonstrated high-fidelity emulation of two-qubit entangled states, we investigated whether states of multiple qubits can also be encoded by our spiking sampling network. Figure 3c shows the fidelity achieved in learning Greenberger-Horne-Zeilinger (GHZ) states [24], i.e. $N$-qubit generalizations of a Bell state, as a function of the number of hidden neurons $M$. The underlying probability distribution covers a larger state space of the visible neurons, requiring us to increase the number of samples to 225000 to reach convergence in the $D_{KL}$. In all cases the fidelity of the learned state to the perfect GHZ state increases with $M$, reaching values of close to 90% and about 70% for three and four qubits, respectively. As layered network architectures are known to require a large number of neurons for representing

GHZ states [15], we assume that larger chip sizes will allow to increase these values further. Note that a GHZ-state fidelity above $\mathcal{F} = 1/\sqrt{2} \approx 70\%$ means that the state exhibits genuine $N$-partite entanglement (cf. dashed line in Fig. 3c) [25].

# 5 Deep and partially restricted networks

Our flexible learning scheme allows the training of network architectures beyond simple bipartite graphs. To explore network architectures with potentially larger representational power we added connections between the visible neurons, resulting in a more densely connected network. Figure 4a shows that a Bell state can be encoded successfully with this architecture, reaching similar fidelities as the two-layer fully restricted spiking network. We also explored deeper network architectures by adding an additional hidden layer, see Fig. 4b. Again, the Bell state was learned successfully reaching similar fidelities as in the bipartite case. We note that the learning performance is not monotonic at small $M$ for $M_2 = 10$ neurons in the second hidden layer. This is expected, since the intermediate layer constitutes an information bottleneck towards the visible layer, which makes learning more difficult. Therefore, the greater representational power offered by additional depth [26] does not necessarily translate into a higher fidelity for $M < M_2$. The overall non-monotonic dependence of the fidelity on the number of hidden neurons is caused by hardware noise leading to fluctuating training performance.

The fact that the learning performance does not improve when using different architectures indicates that the reachable fidelity is currently limited by technical imperfections rather than the representational power of the ansatz. Larger-scale systems may be able to exploit the greater representational power of these deeper and more complex architectures.

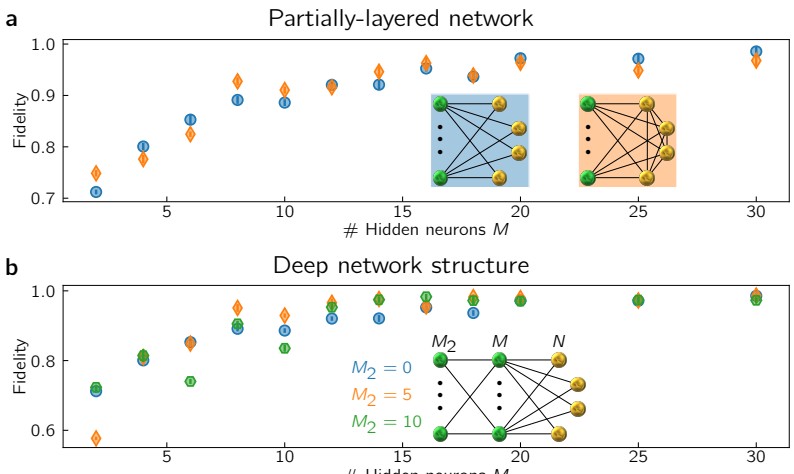

Figure 4: **Extending the network architecture. a**, Fidelity between network-encoded and perfect Bell state, Eq. (5), as a function of the number of hidden neurons for strictly layered network (blue) and an architecture with additional connections between the visible neurons (orange). **b**, Quantum fidelity for states encoded in a deep network architecture with a second hidden layer containing 5 (orange) and 10 (green) neurons compared to the restricted two-layer network (blue).

# 6 Conclusion

We have shown that a spiking neural network implemented on a classical neuromorphic chip can approximate entangled quantum states of few particles with high quantum fidelity. In particular states with non-classical Bell correlations can be encoded faithfully, demonstrating that the representation of quantum states on a classical spiking network can capture their intrinsic quantum features.

The fidelities and system sizes achieved in this first study on neuromorphic quantum state encoding should be regarded as a proof of principle. The experienced restrictions are mainly technical in nature and can be improved in future generations of spiking neuromorphic devices. Specifically for the BrainScaleS-2 system, both the hardware and its surrounding software framework are in an ongoing maturation process. The size and fidelity of the approximated quantum states can be significantly improved upon by optimizing the usage of hardware real-estate, the signal-to-noise ratio of the analog circuitry and the calibration of the chip. Judging from the current pace of progress in neuromorphic engineering, significantly larger systems, both digital and analog, can be expected to become available in the near future [1].

Furthermore, runtime improvements are anticipated, as the current bottleneck is the calculation of the weight updates of the network parameters, which is done "offline" on a conventional computer and only the sampling itself is performed on the chip (see App. A). Using the on-chip plasticity processor to update synaptic weights has the potential of drastically reducing the training time by removing the cumbersome chip-host loop [27].

One key advantage of this neuromorphic system as compared with simulated generative models is that scaling to larger network sizes does not increase the time needed to collect a desired number of samples. We illustrate this property by comparing the sampling time on a neuromorphic chip with sampling times achieved in CPU implementations in App. B showing a gain through neuromorphic sampling already at moderate system sizes. Given the efficient learnability [28] and representability of important classes of quantum states [29–31], and the availability of sampling schemes for neuromorphic devices [32,33], we thus expect favorable scaling properties for our approach. Thus our work opens up a path towards applications of neuromorphic hardware in quantum many-body physics.

## Acknowledgments

We are indebted to the late K. Meier who envisioned and championed the BrainScaleS system and made seminal contributions to this endeavor. We thank A. Kungl for discussions and the Electronic Vision(s) group, in particular E. C. Müller, C. Mauch, Y. Stradmann, P. Spilger, J. Weis, and A. Emmel, for maintaining and providing access to the BrainScaleS-2 system and for technical support.

**Author contributions** S.C. and A.B. carried out the experiment, analysed the data and wrote the paper. S.B., B.C., and A.B. configured the neuromorphic hardware and provided the software interface to access it. S.C., M.G., and T.G. developed the theory and designed the neuromorphic encoding scheme. All authors contributed to interpreting the data and writing the manuscript.

**Funding Information** This work is supported by the Deutsche Forschungsgemeinschaft (DFG, German Research Foundation) under Germany's Excellence Strategy EXC 2181/1 – 390900948 (the Heidelberg STRUCTURES Excellence Cluster), by the DFG – project-ID

273811115 – SFB 1225 (ISOQUANT), by the European Union 7th and Horizon-2020 Framework Programmes, through the ERC Advanced Grant EntangleGen (Project-ID 694561) and under grant agreements 604102, 720270, 785907, 945539 (HBP), and by the Manfred Stärk Foundation.

**Data availability**  The data that support the findings of this study, as well as the code to generate the presented results using the BrainScaleS-2 system, and the scripts to analyze the data are available at https://github.com/sCzischek128/SpikingNeuromorphicChipLearnsEntangled QuantumStates

# A   Implementation details of BrainScaleS-2

The BrainScaleS-2 system is a mixed-signal neuromorphic platform. Its analog core is composed of neuron and synapse circuits with inherent time constants of the order of microseconds. An application-specific integrated circuit (ASIC) for the BrainScaleS-2 system features 512 neuron circuits, which emulate the adaptive exponential integrate-and-fire model. These individual compartments can be wired to resemble more complex structured neurons. An on-chip analog parameter memory as well as integrated static random-access memory (SRAM) cells allow us to individually configure and optimize the dynamics of each circuit. Each neuron integrates input from 256 dedicated synapses, which carry a 6-bit weight and can be either excitatory or inhibitory.

The analog core is accompanied by supporting logic, including circuitry for communication and configuration. Further functionality is provided by high-bandwidth spike sources, which can emit either regular or Poissonian spike trains of configurable frequency. A routing module allows mixing these spikes with external stimuli and recurrent events. It allows, in combination with in-synapse event filtering, the implementation of arbitrary network topologies.

Custom embedded processors allow the modification of the entire configuration space during the runtime of an experiment. Tightly coupled to the synaptic arrays, they allow the efficient and flexible implementation of learning rules based on observables such as neuronal potentials, firing rates, and synaptic correlations.

A network of leaky integrate-and-fire (LIF) neurons can implement a sampling spiking network (SSN) if the neurons are under stochastic noise influence, their membrane time constant is sufficiently small and the synaptic and refractory time constants roughly match [18]. A system-specific calibration is required to configure the analog core of BrainScaleS-2, shown in Fig. 5a according to these requirements. For ease of implementation we use a simple routing scheme in which the on-chip network looks like 128 unique sources which can be arbitrarily connected. This allows the association of each of the 128 synapse drivers with one spike source while using the double line to implement signed synapses (cf. Fig. 5b).

The stochastic input spikes are generated via two of the eight on-chip linear shift registers (LSFRs). We assign the spike source IDs 0-63 to the network neurons and split the spike trains from the LSFRs among the IDs 64-127. For networks smaller than 64 neurons, the upper part of (0-63) remains unused. Again simplifying the implementation we use the first half of the noise IDs (64-95) as excitatory and the second half (96-127) as inhibitory sources (cf. Fig. 5c lower part). This scheme allows in principle all-to-all connectivity within the network. Choosing to use a layered network structure results in a block structure of the upper part of the synapse array (cf. Fig. 5c).

Each sampling neuron is connected to 5 randomly chosen excitatory and 5 randomly chosen inhibitory noise sources. This introduces correlations between neurons even without synaptic connections, but in general does not hinder training [16, 34]. Synaptic connections

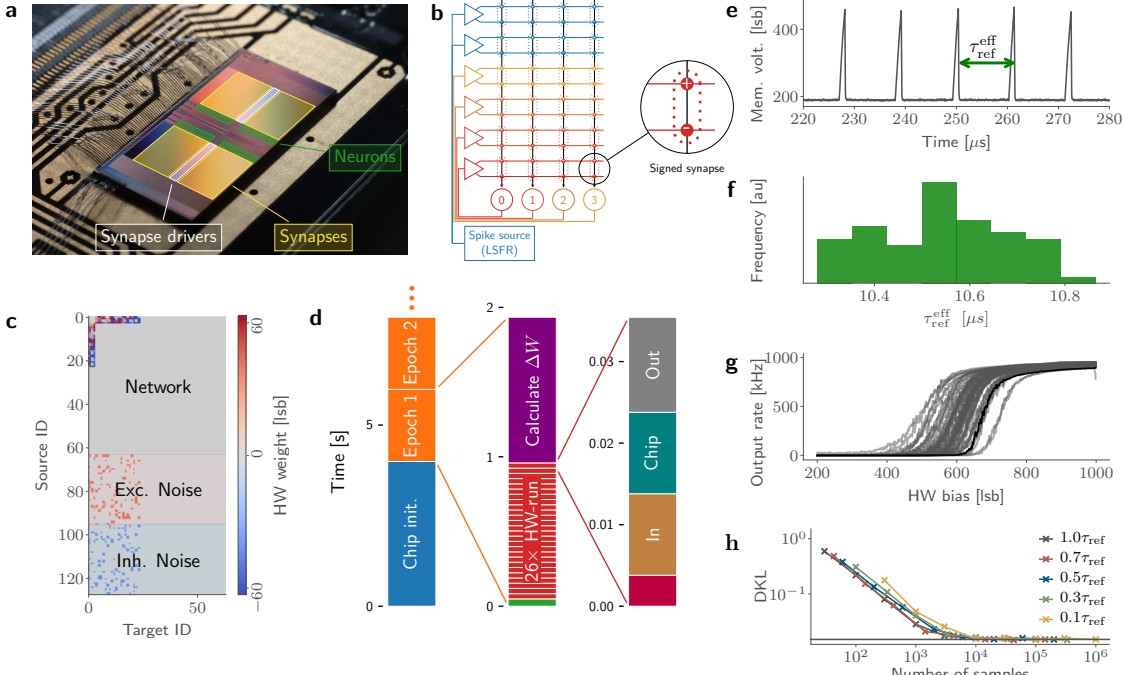

Figure 5: **Details of the BrainScaleS-2 neuromorphic chip.** **a**, Photograph of the BrainScaleS-2 chip with circuits of 4×128 AdEx-LIF neurons (green), 2×2×128 synapse drivers (white) and 4 synapse arrays with 256×128 synapses (yellow). **b**, Routing schematic used to implement the sampling spiking network. Each synapse driver projects to two synapse rows in order to allow signed synapses. **c**, Utilized logical connectivity matrix projecting onto the 64 neurons used. Network (neuron-to-neuron) connections are truncated at index 24 (4 visible and 20 hidden) and intra-layer connections are not used. Each neuron receives noise input from 5 excitatory (64-95) and 5 inhibitory (96-127) sources, generated by one on-chip LSFR each. Each connection selects the appropriate synapse row depending on its sign (cf. **b**) **d**, Time usage across a training experiment. The initial configuration (blue) of the chip is comparable to a single epoch (orange). Each epoch consists of a parameter update (green), 26 sampling runs (red) and the update calculation (purple). Each hardware run consists of the construction of the playback program (ruby), the initial buffering on the FPGA (brown), the actual chip runtime (turquoise) and the readout to the host (grey). **e**, Membrane trace of an exemplary neuron at the high-bias end. $\tau_{\text{ref}}^{\text{eff}}$ is the inter spike interval. **f**, Histogram of measured $\tau_{\text{ref}}^{\text{eff}}$. Variations are due to the analog nature of the system. **g**, Activation functions as a function of the leak potential under noise input of the 64 neurons used. $\tau_{\text{ref}}^{\text{eff}}$ is estimated by the output frequency at the high-bias end. **h**, Sampling performance as a number of samples, rather than execution time for different sampling time deltas $dt$. More than two samples per refractory time $\tau_{\text{ref}}^{\text{eff}} \approx 10\,\mu s$ increase the Kullback-Leibler divergence as the samples are not independent.

on BrainScaleS-2 are 6-bit-valued circuits. The dynamical impact of a single network spike (used to mediate the stochastic response of the receiving sampling unit) onto another neuron is given by its own strength relative to the total strength of the input provided by the background sources. The latter defines the transfer function and thereby the excitability of the neurons (cf. Fig. 5g). Choosing the noise parameters (weight and number of sources) is done such as to attain the competing goals of allowing the network neurons to drive each other significantly while allowing for small weight changes within the 6-bit resolution limit. The particular choice is, in general, problem dependent.

Having chosen the noise parameters, the sampling interface of BrainScaleS-2 becomes a black box that requires a weight (6-bit) matrix and a (10-bit) bias vector and returns a set of spike trains. Neurons are assigned a state of $z = 1$ at time $t$ if they emitted a spike within their effective refractory period $\tau_{\text{ref}}^{\text{eff}}$ prior to $t$ (cf.Fig. 1c in the main text). We determine $\tau_{\text{ref}}^{\text{eff}}$ by setting the leak potential of the neurons to its maximum value and measuring the resulting inter-spike intervals (cf. Fig. 5e). The effective refractory time consists of the clamped part which is digitally driven and therefore does not vary between different neurons and the drift part back to the spiking threshold in the end. Due to the circuit variability (e.g. different membrane time constants) of the analog circuits we see some modest variation in $\tau_{\text{ref}}^{\text{eff}}$ (cf. Fig. 5f). Using the measured $\tau_{\text{ref}}^{\text{eff}}$ we assign a state every $2\,\mu\text{s}$ and use the set of these states for the evaluation and the update calculation.

Figure Fig. 5h demonstrates the correctness of an approximated distribution for a simulated sampling spiking network (using [35]) as a function of the number of samples for different state assign times $dt$ (cf. Fig. 1 in the main text). For more than two samples per refractory period $\tau_{\text{ref}}$ the number of samples required to achieve a given performance level increases due to the correlated states as expected from the Nyquist-Shannon theorem. Both the noise parameters and the sample frequency were chosen such that they enable sufficiently accurate sampling, but without performing an exhaustive optimization.

As discussed above, a chip-specific calibration is required but can be reused for each training. For each experiment the chip needs to be initialized (blue period in Fig. 5d) once. This ensures that the correct calibration is loaded and the routing is configured correctly before the training iterations (orange period in Fig. 5d) can start. After the initialization only the synapse array (weights) and the leak potentials of the neurons (biases) are reconfigured once per epoch (green period in Fig. 5d). Each training epoch consists of 26 sampling runs (red section in Fig. 5d) and a single calculation of the parameter update (purple in Fig. 5d). In each hardware run we build a program for the FPGA to execute (dark red in Fig. 5d), transfer it to the FPGA with some initial buffering (yellow in Fig. 5d) in order to compensate for network latencies, perform the actual execution on chip (light blue in Fig. 5d) and transfer the spikes back to the host computer (grey in Fig. 5d).

In total, an epoch takes about $1.9\,\text{s}$ of which roughly half is spent in the sampling and the other half is used to calculate the parameter updates. While some time was spent to improve performance, both parts can still be optimized. For example the gradient calculation is implemented in Python and most of the sampling time is spent buffering and reading back the results. The actual hardware runtime is only $30\,\%$ of the time marked as *HW-run* in Fig. 5d. Using a more complex routing setup an increase to at least 256-spike sources is possible and since BrainScaleS-2 is a physical system the runtime of the hardware part is not affected by the network size.

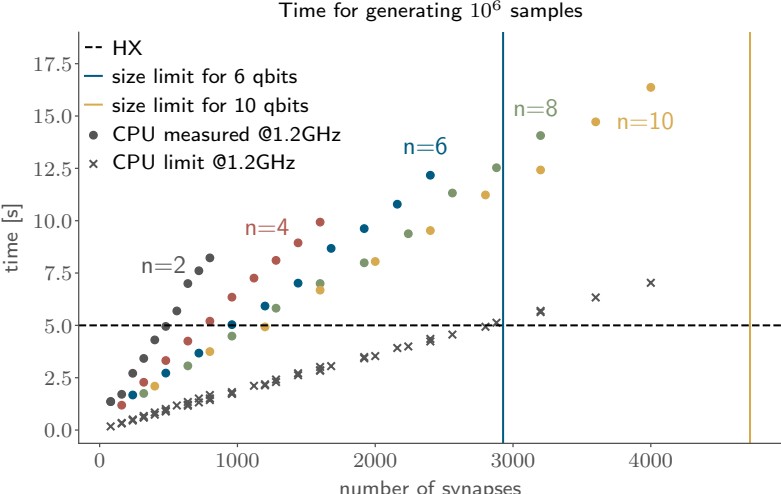

Figure 6: Measured (dots) and estimated (crosses) sampling times for the generation of a million samples, for different quantum system size ($N = 2, 4, 6, 8, 10$ spins, colours) and hidden layer sizes ($M = 20, 40, 60, 80, 100, 120, 140, 160, 180, 200$) on a Intel Xeon E5-2630v4 compared to the constant runtime of the BrainScaleS-2 system (horizontal line). The software time estimation assumes one FLOP per clock cycle and one FLOP required per synaptic interaction, bias and state assignment (see text). The number of neurons on BrainScaleS-2 is limited to $(2N) + M < 256$ which limits the implementable system size (vertical lines, for 6 spins and 244 hidden neurons and 10 spins and 236 hidden neurons).

# B Computation time benchmark for sampling from neural networks

In this section, we provide a speed comparison between the BrainScaleS-2 neuromorphic chip and a C++-implemented software solution to the sampling from binary Boltzmann machines. The software implements standard Gibbs sampling, i.e. it sequentially calculates the "membrane potential" $u_i = b_i + \sum_k W_{ki} z_k$ for each neuron and assigns a new state $z_i = 1$ with probability $\sigma(u_i) = 1/(1 + \exp[-u_i])$ and $z_i = 0$ otherwise. This implementation, while fairly optimized in single-thread performance, does not take into account the potential parallelism of a layered structure. Since the simulator is optimized for large-scale systems it drops all terms with $W_{ki} = 0$, at the price of an additional indirection. The sum now runs over a list of indices which is harder to optimize than a simple sequential iteration. We executed this on the bwForCluster NEMO cluster [36] which uses Intel Xeon E5-2630v4 (Broadwell) CPUs.

Generating a new state requires the update of all neurons, and each update of a single neuron requires the calculation of $u_i$ plus a comparison with a random number for the probabilistic update. For the architecture used in the main manuscript, i.e. layered networks with $2N$ visible and $M$ hidden neurons and assuming a perfect implementation without additional cost for memory accesses, generating a new update takes $2(2N)M$ evaluations and additions of the term $W_{ki} z_k$, besides $(2N) + M$ additions of $b_i$, and $(2N) + M$ comparisons to a random number. Assuming further that each of these steps takes one clock cycle, we can estimate the expected time required.

In order to reduce the impact of the initialization of the software sampler (loading of the network configuration and initialization) we measure the time to generate $10^6$ samples. We note that the number of operations per update is dominated by the number of connections

(synapses) $2(2N)M$. As such, the time required scales linearly in the number of hidden units only for a fixed number of visible units, which is given by the size of the physical system (cf. Fig. 6).

On the other hand, the BrainScaleS-2 implementation, due to its inherently parallel architecture, requires a sample generation time that is independent of the size of the sampled network. With $\tau_{\text{ref}}/2 = 5\,\mu\text{s}$ per sample (cf. Fig. 5h), this leads to a constant time of 5 s. This constant scaling is only true if the network fits onto the system (up to 256 sampling neurons). Since the number of visible neurons is given by the size of the physical system that is represented ($N$ spins), larger physical systems give a greater speedup. Already for the case of 8 spins (16 visible units and 180 hidden units) the fixed runtime of the BrainScaleS-2 system is exceeded by our estimation from the idealized software estimate (cf. Fig. 6). Larger system sizes will skew this comparison further to favor of BrainScaleS-2, which can even implement more densely connected network topologies without incurring a performance penalty. We also note that the BrainScaleS-2 chip requires less than 500 mW [37,38], while the Intel Xeon E5-2630v4 has a thermal design power (TDP) of 85 W for 10 cores. As such BrainScaleS-2 is using comparable energy even for the smallest systems we implemented in the prototype system used in the main manuscript. While the system size at which the BrainScaleS-2 chip outperforms CPU implementations may shift to larger values when comparing to the fastest currently available CPUs, the fundamental difference in scaling behavior, i.e. constant v.s. linear, persists.

## C    Representation of the Bell state

The Bell state, $|\Psi^+\rangle = 1/\sqrt{2}\,(|\uparrow\uparrow\rangle + |\downarrow\downarrow\rangle)$, is described by the density matrix

$$\rho_{\text{B}} = \frac{1}{2} \begin{bmatrix} 1 & 0 & 0 & 1 \\ 0 & 0 & 0 & 0 \\ 0 & 0 & 0 & 0 \\ 1 & 0 & 0 & 1 \end{bmatrix}$$

in the standard basis. To encode this state in a spiking neural network, we map it to a POVM probability distribution.

While several choices of POVM representations are possible, we here focus on the tetrahedral representation, where each measurement projects a single qubit onto one corner of a tetrahedron in the Bloch sphere [15]. The POVM elements $M_{a_i}$ for each qubit $i$ can hence be expressed in the form $M_{a_i} = \left(\mathbb{1} + \boldsymbol{s}_{a_i} \boldsymbol{\sigma}\right)/4$, with Pauli operators $\boldsymbol{\sigma} = (\sigma^x, \sigma^y, \sigma^z)$ and $s_{a_i=0} = (0,0,1)$, $s_{a_i=1} = 1/3\left(2\sqrt{2}, 0, -1\right)$, $s_{a_i=2} = 1/3\left(-\sqrt{2}, \sqrt{6}, -1\right)$, $s_{a_i=3} = 1/3\left(-\sqrt{2}, -\sqrt{6}, -1\right)$. The POVM elements thus take the form

$$M_{a_i=0} = \frac{1}{2} \begin{bmatrix} 1 & 0 \\ 0 & 0 \end{bmatrix}, \quad M_{a_i=1} = \frac{1}{6} \begin{bmatrix} 1 & \sqrt{2} \\ \sqrt{2} & 2 \end{bmatrix},$$
$$M_{a_i=2} = \frac{1}{12} \begin{bmatrix} 2 & -\sqrt{2} - i\sqrt{6} \\ -\sqrt{2} + i\sqrt{6} & 4 \end{bmatrix},$$
$$M_{a_i=3} = \frac{1}{12} \begin{bmatrix} 2 & -\sqrt{2} + i\sqrt{6} \\ -\sqrt{2} - i\sqrt{6} & 4 \end{bmatrix}. \tag{6}$$

With this, the POVM probability distribution of the Bell state, $P_{\text{B}}(a_1, a_2) = \text{Tr}\left[\rho_{\text{B}} M_{a_1} \otimes M_{a_2}\right]$, evaluates to

$$P_{\text{B}} = \frac{1}{8} \begin{bmatrix} 1 & 1/3 & 1/3 & 1/3 \\ 1/3 & 1 & 1/3 & 1/3 \\ 1/3 & 1/3 & 1/3 & 1 \\ 1/3 & 1/3 & 1 & 1/3 \end{bmatrix}, \tag{7}$$

where columns correspond to the index $a_1$ and rows to the index $a_2$.

To reconstruct the density matrix from this probability distribution, the inverse of the full-system overlap matrix $T$ is needed, which can be constructed as the product of the single-qubit overlap matrices, $T = T_1 \otimes T_2$. Each single-qubit overlap matrix consists of the elements $T_{a_i, a_i'} = \text{Tr}\left[M_{a_i} M_{a_i'}\right]$. For the tetrahedral POVM the inverse $T_i^{-1}$ of the single-qubit overlap matrix takes the form

$$T_i^{-1} = \begin{bmatrix} 5 & -1 & -1 & -1 \\ -1 & 5 & -1 & -1 \\ -1 & -1 & 5 & -1 \\ -1 & -1 & -1 & 5 \end{bmatrix}. \tag{8}$$

The density matrix can then be reconstructed linearly as $\rho = \sum_{\{a_1, a_2\}} P(a_1, a_2) \mathcal{Q}_{a_1, a_2}$, with operators $\mathcal{Q}_{a_1, a_2} = \sum_{\{a_1', a_2'\}} (T_{a_1, a_1'}^{-1} \otimes T_{a_2, a_2'}^{-1})(M_{a_1'} \otimes M_{a_2'})$.

Furthermore, expectation values of general operators $\mathcal{O}$ can be rewritten in terms of the probability distribution $P(a_1, a_2)$,

$$\langle \mathcal{O} \rangle = \text{Tr}[\rho \mathcal{O}] = \sum_{\{a_1, a_2\}} Q_{a_1, a_2}^{\mathcal{O}} P(a_1, a_2),$$

with $Q_{a_1, a_2}^{\mathcal{O}} = \sum_{\{a_1', a_2'\}} \text{Tr}\left[\mathcal{O} M_{a_1'} \otimes M_{a_2'}\right] T_{a_1, a_1'}^{-1} \otimes T_{a_2, a_2'}^{-1}$. This enables an efficient evaluation of expectation values by sampling configurations from $P(a_1, a_2)$ in the POVM representation, where the density matrix does not need to be calculated explicitly. The POVM representations of important classes of quantum states can be approximated well and in a scalable way by generative modelling approaches [15]. The computational bottleneck of these methods is the generation of samples from the model distribution, and can potentially be alleviated using neuromorphic devices.

The Bell state is encoded in a sampling spiking network as follows. The visible neurons $\boldsymbol{v}$ are identified with the qubits $\boldsymbol{a}$ in the POVM representation. The network parameters are trained such that the distribution $P_{\text{B}}(a_1, a_2)$ is represented by the network. To achieve this, we need to translate the variables $a_1, a_2$, which can take four possible values each, into binary neurons $\boldsymbol{v}$, where each neuron can take the values 0 or 1. The mapping to four binary visible neurons $v_1, \ldots, v_4$ is accomplished by defining

$$a_1 = 2v_1 + v_2, \quad a_2 = 2v_3 + v_4. \tag{9}$$

From this we can derive the distribution $p_{\text{B}}^*(\boldsymbol{v})$ over the states of the visible neurons and have all ingredients to encode the Bell state in our spiking network.

Analogously, the probability distribution for the two-qubit Werner state with noise contribution $r$ can be derived from its density matrix [23, 39],

$$\rho_{\text{W}} = \frac{1}{4} \begin{bmatrix} 1+r & 0 & 0 & 2r \\ 0 & 1-r & 0 & 0 \\ 0 & 0 & 1-r & 0 \\ 2r & 0 & 0 & 1+r \end{bmatrix}. \tag{10}$$

The same is true for Greenberger-Horne-Zeilinger (GHZ) states of more than two qubits, described by the density matrices [24],

$$\rho_{\text{GHZ}} = \frac{1}{2}\begin{bmatrix} 1 & 0 & \dots & 0 & 1 \\ 0 & & & & 0 \\ \vdots & & \mathbf{0} & & \vdots \\ 0 & & & & 0 \\ 1 & 0 & \dots & 0 & 1 \end{bmatrix}. \tag{11}$$

We can then approximate the corresponding probability distributions by a spiking sampling network.

## D  Training algorithm

Our goal is to approximate a target distribution $p^*(\boldsymbol{v})$ by the model distribution $p(\boldsymbol{v};\mathcal{W})$ encoded by the spiking neuromorphic hardware. The distance between the two distributions is quantified by the Kullback-Leibler divergence,

$$\begin{aligned} D_{\text{KL}}(p^*\|p) &= \sum_{\{\boldsymbol{v}\}} p^*(\boldsymbol{v}) \ln\left[\frac{p^*(\boldsymbol{v})}{p(\boldsymbol{v};\mathcal{W})}\right] \\ &= \sum_{\{\boldsymbol{v}\}} p^*(\boldsymbol{v})\Big(\ln[p^*(\boldsymbol{v})] - \ln[\tilde{p}(\boldsymbol{v};\mathcal{W})] + \ln[Z(\mathcal{W})]\Big). \end{aligned} \tag{12}$$

Here we assumed that $p(\boldsymbol{v};\mathcal{W})$ is well described by the marginal of a Boltzmann distribution and introduced the unnormalized probability distribution $\tilde{p}(\boldsymbol{v};\mathcal{W}) = \sum_{\{\boldsymbol{h}\}}\exp[-E(\boldsymbol{v},\boldsymbol{h};\mathcal{W})]$ as the exponential of the negative network energy, as well as the partition sum $Z(\mathcal{W}) = \sum_{\{\boldsymbol{v},\boldsymbol{h}\}}\exp[-E(\boldsymbol{v},\boldsymbol{h};\mathcal{W})]$, which allows us to replace $p(\boldsymbol{v};\mathcal{W}) = \tilde{p}(\boldsymbol{v};\mathcal{W})/Z(\mathcal{W})$ in the second line of Eq. (12).

The gradient of the Kullback-Leibler divergence with respect to a general connecting weight $W_{i,j}$ is given by

$$\begin{aligned} \frac{\partial D_{\text{KL}}}{\partial W_{i,j}} &= \sum_{\{\boldsymbol{v}\}} p^*(\boldsymbol{v})\left[-\frac{1}{\tilde{p}(\boldsymbol{v};\mathcal{W})}\frac{\partial\tilde{p}(\boldsymbol{v};\mathcal{W})}{\partial W_{i,j}} + \frac{1}{Z(\mathcal{W})}\frac{\partial Z(\mathcal{W})}{\partial W_{i,j}}\right] \\ &= \sum_{\{\boldsymbol{v}\}} p^*(\boldsymbol{v})\left[-\frac{1}{\tilde{p}(\boldsymbol{v};\mathcal{W})}\left(\sum_{\{\boldsymbol{h}\}} v_i h_j e^{-E(\boldsymbol{v},\boldsymbol{h};\mathcal{W})}\right) + \frac{1}{Z(\mathcal{W})}\left(\sum_{\{\boldsymbol{v}',\boldsymbol{h}\}} v_i' h_j e^{-E(\boldsymbol{v}',\boldsymbol{h};\mathcal{W})}\right)\right] \\ &= -\sum_{\{\boldsymbol{v},\boldsymbol{h}\}}\frac{p^*(\boldsymbol{v})}{p(\boldsymbol{v};\mathcal{W})} v_i h_j \frac{e^{-E(\boldsymbol{v},\boldsymbol{h};\mathcal{W})}}{Z(\mathcal{W})} + \sum_{\{\boldsymbol{v}',\boldsymbol{h}\}} v_i' h_j \frac{e^{-E(\boldsymbol{v}',\boldsymbol{h};\mathcal{W})}}{Z(\mathcal{W})} \\ &= \sum_{\{\boldsymbol{v},\boldsymbol{h}\}}\left[1 - \frac{p^*(\boldsymbol{v})}{p(\boldsymbol{v};\mathcal{W})}\right] v_i h_j \frac{\exp[-E(\boldsymbol{v},\boldsymbol{h};\mathcal{W})]}{Z(\mathcal{W})} \\ &= \left\langle\left[1 - \frac{p^*(\boldsymbol{v})}{p(\boldsymbol{v};\mathcal{W})}\right] v_i h_j\right\rangle_{p(\boldsymbol{v},\boldsymbol{h};\mathcal{W})}. \end{aligned} \tag{13}$$

Thus, the weight updates are calculated by drawing a sample set of network states, evaluating the probability $p(\boldsymbol{v};\mathcal{W})$ underlying the configurations in the set, and calculating the expectation value of the product of the two connected neurons, weighted with $1 - p^*(\boldsymbol{v})/p(\boldsymbol{v};\mathcal{W})$. When using the spiking network on the BrainScaleS-2 system, we draw these sample states by

observing the network at regular points in time spaced by $2\,\mu s$ (for a refractory time of about $10\,\mu s$, see App. A).

The weight update in training epoch $t$ then reads

$$W_{i,j}^t = W_{i,j}^{t-1} - \eta \left\langle \left[1 - \frac{p^*(\boldsymbol{v})}{p(\boldsymbol{v};\mathcal{W})}\right] v_i h_j \right\rangle_{p(\boldsymbol{v},\boldsymbol{h};\mathcal{W})} , \tag{14}$$

with learning rate $\eta$. Analogously, updates for the biases can be derived,

$$b_j^t = b_j^{t-1} - \eta \left\langle \left[1 - \frac{p^*(\boldsymbol{v})}{p(\boldsymbol{v};\mathcal{W})}\right] h_j \right\rangle_{p(\boldsymbol{v},\boldsymbol{h};\mathcal{W})} ,$$

$$d_i^t = d_i^{t-1} - \eta \left\langle \left[1 - \frac{p^*(\boldsymbol{v})}{p(\boldsymbol{v};\mathcal{W})}\right] v_i \right\rangle_{p(\boldsymbol{v},\boldsymbol{h};\mathcal{W})} . \tag{15}$$

If connections between the visible neurons exist in the network structure, the updates for those connecting weights are analogous to Eq. (14), where the weighted expectation value of the product of the corresponding visible neurons is evaluated. This learning scheme is a modified version of wake-sleep learning [13].

Since this training algorithm is based on a gradient-descent ansatz, we can apply further modifications which lead to better convergence, such as a momentum approach to avoid getting stuck at a local minimum. In our simulations, we apply the Adam optimizer scheme. This scheme combines a momentum approach with an adaptive learning rate which is chosen for each network parameter individually. The update for a general network parameter $\mathcal{W}_k$ is given, in the Adam optimizer, by

$$m_k^t = \beta_1 m_k^{t-1} + (1-\beta_1) \frac{\partial D_{\mathrm{KL}}(p^* \| p)}{\partial \mathcal{W}_k} , \quad v_k^t = \beta_2 v_k^{t-1} + (1-\beta_2) \left[\frac{\partial D_{\mathrm{KL}}(p^* \| p)}{\partial \mathcal{W}_k}\right]^2 ,$$

$$\hat{m}_k^t = \frac{m_k^t}{1-\beta_1^t} , \quad \hat{v}_k^t = \frac{v_k^t}{1-\beta_2^t} ,$$

$$\mathcal{W}_k^t = \mathcal{W}_k^{t-1} - \eta \frac{\hat{m}_k^t}{\sqrt{\hat{v}_k^t} + \varepsilon} , \tag{16}$$

where $m_k$ acts as a momentum and $v_k$ sets the adaptive learning rate. Here we follow the common choice and set the hyper-parameters to $\beta_1 = 0.9$, $\beta_2 = 0.999$, and $\varepsilon = 10^{-8}$, [40]. We additionally multiply the adaptive learning rate with an exponentially decaying factor $\eta(t)$ from an initial value of $\eta_{\mathrm{init}} = 1$ to a minimum value of $\eta_{\mathrm{min}} = 0.001$,

$$\eta(t) = \max(\eta_{\mathrm{init}} \exp[-0.001t], \eta_{\mathrm{min}}) , \tag{17}$$

where $t$ counts the training epochs. Note that this learning rate is a hyper-parameter that needs to be chosen accordingly and requires a special form for the discrete-valued weights and biases on the neuromorphic hardware. With the exponentially decaying factor we ensure that the learning rate is large enough to cause changes in the weights at short time scales, but is small enough to enable convergence at later times.

In general, Hebbian training algorithms are based on minimizing the correlation mismatch between data and model distributions. The traditional way for estimating this mismatch is contrastive divergence [13, 41], where the target and model distributions are approximated by a single layer-wise network update (CD-1). However, the improved performance of contrastive divergence relies on the fact that preparing the software network in a defined state is cheap compared to calculating updates of neuron configurations. Our neuromorphic hardware, as a physical dynamical system, implicitly calculates the neuron updates and the actual sampling run is cheap compared to the cost of the network initialization with the given

performance characteristics. An implementation of the preparation-dominated contrastive divergence scheme on the spiking neuromorphic hardware hence does not provide any of the benefits observed in software simulations. In contrast we take advantage of these hardware characteristics by using the full model distribution to calculate network parameter updates, which improves the quality of the stochastic gradient estimation. We further optimize the hardware training implementation by reconstructing the correlations between visible and hidden layers from the encoded distribution by reweighting all samples $p(\nu)$ according to the target probability $p^*(\nu)$, see Eq. (14) and Eq. (15). This is in contrast to contrastive divergence learning where the distribution of the visible layer is explicitly enforced to match the target distribution and only the correlations with the hidden layer are being sampled. Beyond the optimized implementation on the spiking neuromorphic system, our proposed training algorithm can be used to obtain network parameter updates for arbitrarily connected networks, while contrastive divergence is limited to strictly layered network structures.

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
