# Peer review of "Spiking neuromorphic chip learns entangled quantum states"

_SciPost Physics, doi:SciPost Phys. 12, 039 (2022)_

## Round 3 · Referee Report · Anonymous (Referee 1) · 2021-3-19

Strengths

Interdisciplinary work connecting neuromorphic chips and representations of quantum states.

Weaknesses

Easily accessible explanation of the functionality of the neurmorphic chip missing.

Report

The paper presents an approach to represent quantum states with the help of a neural network that is implemented in a neuromorphic chip in hardware. The concept shares some similarities with numerical approaches that use software coded neural networks for this applications, see e.g. [29,30], but uses a neural network that is implemented in hardware.

There are some aspects that should be clarified better.

1) As I understand it, the expansion coefficients of a quantum state in a chosen bases are here represented by neurons of the neural network. What is not clear to me is how phases of the coefficients are taken into account. The expansion coefficients are generally complex numbers. However for a physical implementation of the network I would expect real coefficients. is this expectation too naive? How is this taken care of? In this context I also not a misleading (even wrong) statement in the introduction "any quantum state can be mapped to a probability distribution". This is in conflict with the notion that such mapping can't hold for non-classical states, where e.g. the Wigner function becomes negative. An example are Fock states.

2) The connections and differences to approaches with software coded restricted Boltzmann machines as discussed in refs [29,30] should be discussed in more detail. In this context I also note that the software coded restricted Boltzmann machines have also been developed for mixed states as are considered here. I also note that the network isn't called restricted Boltzmann machine here although this is the usual name for it.

3) My understanding is that the employed chip works purely classically. This should be stated explicitly.

4) In addition, for the interdisciplinary nature of the work, an explanation of the employed chip addressed to laymen would be highly appreciated.

---

## Round 3 · Referee Report · Anonymous (Referee 2) · 2021-7-5

Strengths

1) Relevant first step in the simulation of quantum states using neuromorphic hardware.

2) Very clear and exhaustive presentation of the results.

Weaknesses

1) The statements on the future developments of neuromorphic hardware are not always clearly justified.

Report

I have read the paper with interest. The idea is innovative and the overall implementation and description of the experiment is commendable. I am certainly inclined to recommend the paper for publication on SciPost Physics. Still, I have some doubts regarding the comparison between the hardware and the software implementations of the algorithm. In section 2 the following statement is made regarding the training algorithm:

"This otherwise prohibitively compute-intensive method was made possible by the accelerated hardware dynamics and allows a much better approximation of the DKL gradient than the more conventional contrastive divergence update scheme"

I find this statement rather vague as no quantitative comparison between the hardware and software implementations is made in this perspective. A somehow more detailed discussion of this issue is given in Appendix D, but I still find this discussion insufficient. As far as I understood the authors claim that the hardware allows to optimally sample from the complete distributions and, therefore, the sampling does not need to be approximated as in the conventional contrastive divergence approach.

However, when an explicit comparison between the hardware and software implementations is presented (in Appendix B) no comment is made on the relevance of such approximation (which I believe is not made in the software implementation discussed in Sec. B) . I believe that the paper will benefit by a more extensive discussion on the relevance of the contrastive divergence approximation and on the benefit granted by the possibility of avoiding it in the hardware realisation.

Requested changes

1) Extend the discussion on the contrastive divergence approximation and connect this discussion with the specific comparison between hardware and software implementations made in Appendix B.

---

## Round 4 · Referee Report · Anonymous (Referee 2) · 2021-9-7

Report

I am satisfied with the authors reply to my comments and I have no further recommendations. I believe that the article meets the expectations and criteria for publication on SciPost Physics.

---

## Round 4 · Referee Report · Anonymous (Referee 1) · 2021-10-21

Strengths

The work links research areas that developed independently so far. This the characterization of quantum states and the representing neural networks on neuromorphic chips

Weaknesses

A particular weakness is that the description of the result is not as explicit as it should. There are claims that the work is "demonstrating that intrinsic quantum features can be captured by a classical spiking network". Yet, my understanding is that the classical spiking network just learns two distributions, the real and imaginary parts of the matrix element of the density matrix.

Report

This is an interesting work as it explores applications of neuromorphic chips in representing quantum states. A particular weakness of the paper is however that the description of the result is not as explicit as it should. There are claims that the work is "demonstrating that intrinsic quantum features can be captured by a classical spiking network". Yet, my understanding is that the classical spiking network just learns two distributions, the real and imaginary parts of the matrix element of the density matrix. It is no surprise at all that a classical computing device can compute representations of quantum states, even if they are highly correlated and non-classical. Every numerical simulation of a quantum system does this.

Requested changes

I suggest that the presentation of the material should be changed to clearly say that the network is able to approximate the matrix elements of the density matrix representing certain quantum states.
Any statements that seem to suggest that a classical device can show quantum correlations should be removed. I understand that such things are not explicitly claimed, but it is also not stated clearly enough that this claims are not made.

---

## Round 4 · Author Response

We thank the editor for considering our manuscript and for sending us the very constructive reports of the referees. We are convinced that we can resolve the referees' criticism in our response and hope to meet the publication criteria of SciPost Physics with our revised manuscript. Below we provide a point-by-point response to the referees' comments.

Reply to Report 1

We thank the referee for the detailed consideration of our manuscript and the constructive feedback, which we would like to answer point by point in the following.

  • Reply to 1):
    We thank the referee for this interesting comment. Our approach in the manuscript is based on a probabilistic formulation of quantum mechanics, in which a quantum state is represented by the probability distribution over the outcomes of a tomographically complete measurement. Formally, measurements in quantum mechanics are described by sets {M_a} of positive operators (in our case projection operators). This allows us to map a quantum state represented by the density matrix ρ to a positive definite probability distribution via P(a)=tr(M_a ρ). Under certain conditions (see reference [15] in the manuscript), this mapping is invertible, so by knowing P(a) one can reconstruct ρ. In order to make it represent the quantum state ρ, we configure our neuromorphic chip to sample from the probability distribution P(a), which entails the complete information contained in the quantum state. The fact that any density matrix (or wave function, in the case of pure states) can be represented by a probability distribution is indeed crucial for our approach. In the context of phase-space representations, the equivalent of the employed positive operator-valued measure (POVM) is the Husimi distribution, which is defined in terms of projections of the state onto a set of coherent states. In this case the coherent-state projectors constitute the set {M_a}. The Husimi distribution is non-negative and gives a complete representation of the state.

We have added a sentence in the last paragraph on page 4 to clarify this for the reader:

“Hence, any density matrix ρ can be mapped to a probability distribution P(a), and the information contained in the quantum state can be retrieved from that distribution.“

  • Reply to 2):
    We thank the referee for pointing out this ambiguity. It is correct that software-coded restricted Boltzmann machines (RBMs) have been used successfully for encoding quantum many-body states (Ref. [30] in the manuscript provides an overview). One of the motivations of our work is in fact to leverage the fast and energy-efficient sampling capabilities of the neuromorphic chip for eventually speeding up such approaches.

We would like to stress, however, that the neuromorphic hardware should not be considered as a mere emulator of a Boltzmann machine. Most importantly, it is a dynamical system, which, in a non-trivial and approximative manner, develops spike-time correlations that with the chosen network parameterization allow us to approximate a Boltzmann distribution. For that reason, we prefer to avoid the term “Restricted Boltzmann machine”.

We hence rather prefer not to consider our approach as equivalent to variational methods in quantum many-body physics using artificial neural networks, despite the fact that these approaches certainly provide motivation to expect scalability for certain classes of states. The hardware model shows the approximative behavior of an RBM, but in general no exact closed-form expression for the complete distribution underlying the neuron dynamics can be given.

Concerning the representation of mixed quantum states by means of RBMs, we remark that purifications of density operators have been demonstrated [G. Torlai, R. Melko, Latent Space Purification via Neural Density Operators, PRL, 2018; M.J. Hartmann, G. Carleo, Neural-Network Approach to Dissipative Quantum Many-Body Dynamics, PRL, 2019] which result in complex-valued wave functions that cannot directly be translated to our sample-based neuromorphic approach. The probabilistic approach used here has been pioneered by Carrasquilla et al. (see Ref. [15] in the manuscript), who used RBMs as well as more complex network architectures.

We added a sentence after Eq. (3) in the revised manuscript, emphasizing that the spiking neural network does not emulate a restricted Boltzmann machine and hence is different from software encodings of artificial neural networks representing quantum many-body systems:

“While the dynamical behavior of the spiking hardware approximates this probability distribution, the exact relation between the network parameters and the encoded distribution cannot be given in a closed form [18].”

  • Reply to 3):
    We agree with the referee and now explicitly state in the last sentence of the introduction and in the first sentence of the conclusion that the spiking network is classical.

  • Reply to 4):
    In order to extend the discussion of the operation mode of the neuromorphic chip and of the spike-based sampling framework, we reworked the second paragraph on page 4 in the main text.

Reply to Report 2

We thank the referee for the detailed consideration of our manuscript and the constructive feedback, which we would like to answer point by point in the following.

It is correct that the hardware can generate exact samples, which allows us to go beyond the contrastive divergence scheme. The statement in the main text addresses the issue that the wake-sleep learning we employ in classical machine learning suffers from the problem of requiring a large number of samples. Contrastive divergence instead estimates the gradient based on a single sample. For the neuromorphic hardware, the sample generation is not the bottleneck and thus it is feasible to employ wake-sleep learning directly, which is advantageous as compared to contrastive divergence because it increases the quality of the approximative determination of observables.

In Appendix B we concentrate on comparing runtimes for the task of sampling from a given network, independent of the learning algorithm which this sampling is employed for. In the revised manuscript we emphasize in the last paragraph of the conclusion section that the comparison only refers to sampling neuron configurations rather than to the training of the network.

We reformulated and extended the statement quoted by the referee, in the last paragraph of Section 2, in order to make it clearer for the reader, by:

“The performance characteristics of the neuromorphic hardware make computing additional samples cheap compared to reconfiguration and reinitialization. Hence we can take into account the complete sampled distribution for the update calculation, rather than relying on few-sample approximations as in contrastive divergence [13]. This enables a much better estimation of the D_KL gradient and does not rely on layer-wise conditional independence, allowing the exploration of network topologies other than bipartite graphs. See App. D for an extended discussion of the learning scheme.”

In addition, we reworked the last paragraph of Appendix D on pages 18 and 19. We extended this paragraph with a discussion of contrastive divergence, its benefits for software simulations, and its disadvantages when used for accelerated neuromorphic hardware.

---

## Round 4 · List of Changes

• Added "classical" in the last sentence of Section 1 (page 2): "With this substrate, we demonstrate an approximate representation of quantum states with classical spiking neural networks that is sufficiently precise for encoding genuine quantum correlations. "

  • Rewrote the second paragraph of Section 2 (page 4): "The BrainScaleS-2 system, depicted in Fig. 1b, features 512 LIF neuron circuits interacting through a configurable weight matrix [11]. Similar to biological neurons in the human brain, LIF neurons communicate via spikes. Each neuron can be viewed as a capacitor integrating the currents it receives from its synaptic inputs to generate a membrane potential. Whenever this membrane potential crosses a threshold from below, the neuron sends a spike to the synaptic inputs of its efferent partners (Fig. 1c, top panel). After sending a spike, the neuron is set to an inactive state, in which no additional spike can be triggered for a certain time, referred to as the refractory period τ_ref. In the spike-based sampling framework, neurons in this refractory state encode the state z = 1, and z = 0 at all other times (Fig. 1c, lower panel). The stochasticity required for sampling is induced by adding a random component to the generation of spikes; for LIF networks, this can be ensured by sufficiently noisy membrane potentials [16,18]. To this end, we used on-chip sources to inject pseudo-Poisson spike trains into the network (see App. A)."

  • Added a sentence in the fourth paragraph of Section 2 (page 4): "Hence, any density matrix ρ can be mapped to a probability distribution P(a), and the information contained in the quantum state can be retrieved from that distribution. "

  • Rewrote and extended the last paragraph of Section2, after Eq. (3) (page 5): "In each training epoch, the synaptic weights were updated along the gradient of the D_KL (see App. D), which is derived assuming that the distribution p(v;W) is given by Eq.(1). While the dy- namical behavior of the spiking hardware approximates this probability distribution, the exact relation between the network parameters and the encoded distribution cannot be given in a closed form [18]. Instead, pairwise correlations ⟨v_i h_j⟩_{model} in the network were measured from the sampled distribution p(v,h;W). "

  • Added "classical" in the first sentence of Section 6 (page 9): "We have shown that a spiking neural network implemented on a classical neuromorphic chip can approximate entangled quantum states of few particles with high quantum fidelity. "

  • Rewrote the second sentence of the last paragraph in Section 6 (page 9): "We illustrate this property by comparing the sampling time on a neuromorphic chip with sampling times achieved in CPU implementations in App. B showing a gain through neuromorphic sampling already at moderate system sizes."

  • Rewrote and extended the last paragraph of Appendix D (pages 18 and 19): "In general, Hebbian training algorithms are based on minimizing the correlation mismatch between data and model distributions. The traditional way for estimating this mismatch is contrastive divergence [13,41], where the target and model distributions are approximated by a single layer-wise network update (CD-1). However, the improved performance of contrastive divergence relies on the fact that preparing the software network in a defined state is cheap compared to calculating updates of neuron configurations. Our neuromorphic hardware, as a physical dynamical system, implicitly calculates the neuron updates and the actual sampling run is cheap compared to the cost of the network initialization with the given performance characteristics. An implementation of the preparation-dominated contrastive divergence scheme on the spiking neuromorphic hardware hence does not provide any of the benefits observed in software simulations. In contrast we take advantage of these hardware characteristics by using the full model distribution to calculate network parameter updates, which improves the quality of the stochastic gradient estimation. We further optimize the hardware training implementation by reconstructing the correlations between visible and hidden layers from the encoded distribution by reweighting all samples p(v) according to the target probability p^∗(v), see Eq.(15) and Eq.(16). This is in contrast to contrastive divergence learning where the distribution of the visible layer is explicitly enforced to match the target distribution and only the correlations with the hidden layer are being sampled. Beyond the optimized implementation on the spiking neuromorphic system, our proposed training algorithm can be used to obtain network parameter updates for arbitrarily connected networks, while contrastive divergence is limited to strictly layered network structures. "

---

## Round 5 · Author Response

We thank the editor and the referees for considering our revised manuscript. In our revised manuscript, which we resubmit herewith, we have addressed the concerns of the second referee. We are convinced that our paper now meets the acceptance criteria of SciPost Physics. Below we provide a point-by-point response to the referee reports.

*Reply to Report 1*

We thank the referee for his positive feedback and for suggesting the publication of our manuscript.

*Reply to Report 2*

We thank the referee for the positive evaluation of our manuscript and for pointing out the ambiguity which could be seen in the statement (on p. 9) that our work is “demonstrating that intrinsic quantum features can be captured by a classical spiking network”. We indeed do not mean to imply that the classical neuromorphic system is subject to genuine quantum correlations itself. To make clearer that the hardware rather encodes entangled quantum states, we rewrote the sentence cited by the referee, as well as similar formulations in the text, as detailed in the list of changes.

---

## Round 5 · List of Changes

- Last sentence of the abstract, p. 1:
Replace “Extracted Bell correlations for pure and mixed two-qubit states convey that non-classical features are captured by the analog hardware, …” by “Bell correlations of pure and mixed two-qubit states are well captured by the analog hardware, …”

- Last sentence of Section 1, p. 2:
Replace “With this substrate, we demonstrate an approximate representation of quantum states with classical spiking neural networks that is sufficiently precise for encoding genuine quantum correlations.” by “With this substrate, we demonstrate an approximate representation of quantum states with classical spiking neural networks that is sufficiently precise for encoding states with genuine quantum correlations.”

- Third paragraph of Section 3, p. 6:
Replace “The correlations encoded by the trained spiking network clearly exceed the classicality bound …” by “The correlations in the quantum states encoded as probability distributions by the trained spiking network clearly exceed the classicality bound …”

- First paragraph of Section 6, p. 9:
Replace “In particular non-classical Bell correlations can be encoded faithfully, demonstrating that intrinsic quantum features can be captured by a classical spiking network.” by “In particular states with non-classical Bell correlations can be encoded faithfully, demonstrating that the representation of quantum states on a classical spiking network can capture their intrinsic quantum features.”

---

## Editorial Decision

published